# MEC Computation Offloading-Based Learning Strategy in Ultra-Dense Networks

**Chunhong Duo [1,2,*], Peng Dong [1] , Qize Gao [1], Baogang Li [3,4] and Yongqian Li [3,4]**

1 Department of Computer, North China Electric Power University, Baoding 071003, China; dongpeng9624@163.com (P.D.); gaoqize@ncepu.edu.cn (Q.G.)
2 Engineering Research Center of Intelligent Computing for Complex Energy Systems, Ministry of Education, Baoding 071003, China
3 Department of Electronic and Communication Engineering, North China Electric Power University, Baoding 071003, China; baogangli@ncepu.edu.cn (B.L.); 52151708@ncepu.edu.cn (Y.L.)
4 Hebei Key Laboratory of Power Internet of Things Technology, North China Electric Power University, Baoding 071003, China
* Correspondence: duochunhong@163.com

**Abstract:** Mobile edge computing (MEC) has the potential to realize intensive applications in 5G networks. Through migrating intensive tasks to edge servers, MEC can expand the computing power of wireless networks. Fifth generation networks need to meet service requirements, such as wide coverage, high capacity, low latency and low power consumption. Therefore, the network architecture of MEC combined with ultra-dense networks (UDNs) will become a typical model in the future. This paper designs a MEC architecture in a UDN, which is our research background. First, the system model is established in the UDN, and the optimization problems is proposed. Second, the action classification (AC) algorithm is utilized to filter the effective action in Q-learning. Then, the optimal computation offloading strategy and resource allocation scheme are obtained using a deep reinforcement learning-based AC algorithm, which is known as the DQN-AC algorithm. Finally, the simulation experiments show that the proposed DQN-AC algorithm can effectively reduce the system weighted cost compared with the full local computation algorithm, full offloading computation algorithm and Q-learning algorithm.

**Keywords:** mobile edge computing; ultra-dense network; computation offloading; deep reinforcement learning

## 1. Introduction

In recent years, with the vigorous development of mobile internet and pervasive computation, the number of mobile users has increased rapidly, and an increasing number of users select compute-intensive applications [1]. At present, 5G networks are rising to support the massive connections between humans, machines and various services. The rapid development of new application fields, such as interactive games, augmented reality, virtual reality, driverless cars and smart grid, in 5G networks requires stronger computing power and higher energy efficiency [2]. For users, they pose higher requirements for indicators, such as computation latency, energy consumption and the number of equipment connections [3]. Mobile user devices (MUDs), such as mobile phones, laptops and tablets, have limited battery power and computing capacity. It is difficult for MUDs to meet these requirements, and they may not be able to process a large number of applications in a short time. More importantly, due to their limited battery power, it has become an obstacle for MUDs to provide higher requirements. Similarly, they cannot meet the needs of ultra-low delay, ultra-low energy consumption and high reliability in 5G scenarios [4].

To solve this problem, a feasible solution is to offload these compute-intensive tasks to the remote centralized cloud, which provides computing power and storage resources [5].

The task of uploading to the cloud requires more computation time, which leads to a longer delay. Mobile edge computing (MEC) deploys computing and storage resources at the edge of mobile networks, and provides service environments and cloud computing capabilities for MUDs with ultra-low latency and high bandwidth [6]. As one of the key technologies in MEC, computation offloading sends all or some of the computing tasks to MEC servers through a wireless channel, to solve the deficiencies of MUDs in resource storage, computing performance and energy efficiency [7].

Ultra-dense network (UDN) technology achieves a hundredfold increase in system capacity in local hot spots [8]. Through deploying more low-power small base stations in UDNs, space multiplexing is improved, and end-to-end transmission delay is reduced. UDNs can also reduce the pressure of large-scale mobile connections, thus improving the overall performance of the network [9]. At present, the UDN is becoming an effective solution to improve data traffic by up to 1000 times and user experience rate by up to 10 to 100 times in 5G networks [10], and UDNs combined with MEC network architecture will become a typical model of wireless networks [11].

In a MEC architecture-based UDN, MUDs are covered by multiple small base stations, and they can access multiple MEC servers at the same time. MEC servers have different resources and transmission environments, which leads to the problem of resource competition and MEC server selection [12]. Therefore, it is of great significance to select the offloading MEC server and allocate computing resources to meet the service requirements.

This paper studies the problems of MEC computation offloading and resource allocation in the UDN scenario. Considering the impact of computing resource and task delay constraints on system performance, the joint optimization of computation offloading and resource allocation is analyzed and discussed. The main contributions are as follows:

(1)    A MEC architecture-based UDN is designed, and a task weighted cost model based on execution delay and energy consumption is established. The task offloading and resource allocation are combined into an NP-hard optimization problem.

(2)    An action classification (AC) algorithm is developed to select the most suitable edge server, which can reduce the possible values of offloading decisions and improve learning efficiency.

(3)    A deep Q network-based AC algorithm (DQN-AC) is proposed to solve the task offloading and resource allocation problems. First, according to the execution delay and computing resource constraints, the AC algorithm is adopted to select effective actions. Then, the DQN algorithm is used to solve the optimization problem, and the optimal task offloading and resource allocation scheme is obtained through finite iterations.

The remainder of this paper is organized as follows. We review the related work in Section 2. Section 3 describes the system model and problem formulation. In Section 4, the proposed DQN learning strategy with action classification is described in detail. Section 5 presents extensive simulation experiments and results to evaluate the performance of the DQN-AC. Finally, Section 6 concludes this paper.

## 2. Related Work

Researchers have proposed some computation offloading and resource allocation schemes for different optimization objectives. In general, relevant research has been carried out with three goals: to reduce execution delay, to reduce energy consumption and to reduce the total cost of weighed time delay and energy consumption. **To reduce execution delay**, in Ref. [13], the problem of mobile user equipment performing offloading calculations in computing tasks is studied, and the Markov process method is adopted to deal with the problem. Online task offloading methods aiming at optimizing delay are proposed. In Ref. [14], the low complexity online algorithm optimized by Lyapunov is used to solve the offloading decision problem. In Ref. [15], an online computation offloading mechanism is designed to minimize the average expected execution delay of tasks under the constraint of average energy consumption in the moving edge computing system. In [16], how to quickly

solve the problem of task offloading decisions and resource allocation joint optimization in channel coherence time is studied, and an online offloading algorithm based on deep reinforcement learning is proposed. Simulation results show that the three algorithms can effectively reduce the execution delay. In Ref. [17], the application scenarios of the Internet of Things is studied, and execution latency is reduced by reasonably allocating computing resources for computing tasks, and a complete polynomial-time approximation scheme is proposed. In Ref. [18], the offloading calculation problem of delay sensitivity of the Internet of Things is studied, and an iterative heuristic algorithm is proposed to dynamically allocate resources. In Ref. [19], computing offloading and resource allocation are addressed in multiple mobile user systems. In Ref. [20], the application of MEC technology in regional distribution networks is considered, and an asynchronous dominant participant-critic algorithm is proposed.

The above literature all focuses offloading decisions to reduce time delay. However, the battery capacity of MUDs is low, which may influence the offloading policy. Researchers continue to explore and study offloading decisions to **minimize energy consumption**. In Ref. [21], a D2D-assisted MEC system is considered to improve equipment energy efficiency. In Ref. [22], the computation offloading problem of a MEC system based on a cooperative multi-carrier relay is studied, and an efficient energy consumption optimization algorithm is proposed. In Ref. [23], the computing offloading problem of moving edge computing in the scenario of the Internet of Vehicles is studied, and a game algorithm based on a deep Q network is proposed. Additionally, deep reinforcement learning is adopted to minimize the energy consumption of MEC systems. In Ref. [24], the joint optimization of computing offloading and resource allocation in multi-user dynamic MEC systems is studied, and a dual depth Q network algorithm is proposed to minimize the energy consumption of MEC systems. In Ref. [25], the computational offloading of IoT devices in a dynamic MEC system composed of multiple edge servers is studied, and an end-to-end deep reinforcement learning method is proposed.

There are also many studies on how to develop computation offloading strategies that **balance execution delay and energy consumption**. In Ref. [26], the offloading scheduling problem of multiple independent tasks in the MEC system is studied, and a low complexity suboptimal algorithm with alternating minimization is proposed. In Ref. [27], computing offloading and resource allocation in multi-user and multi-task scenarios are addressed. In Ref. [28], the computation offloading problem of multi-user MEC systems is studied, and [28,29] take the combined time delay and the weighted sum cost of energy consumption as the optimization objectives. The offloading strategy based on the deep reinforcement learning algorithm is utilized. In Ref. [30], the MEC network for the intelligent Internet of Things is considered, and the offloading decision is automatically learned by the DQN algorithm to optimize the system performance. Task offloading and cache integration are described as a nonlinear problem in Ref. [31], which is solved by the Q-learning and DQN algorithms. An optimization framework of wireless MEC resource allocation based on reinforcement learning was proposed in Ref. [32], and Q-learning and DQN algorithms were, respectively, used in simulation experiments. In contrast to the single-channel MEC system in Ref. [32], multi-user and multi-channel MEC systems are designed in Ref. [33].

A comparative analysis of the previous work is illustrated in Table 1. In general, most of these works focused on the offloading problem in MEC. In this paper, we focus on a MEC scenario in a UDN, where heterogeneous computational tasks with random arrivals require scheduling to different edge servers with varying delay constraints, and propose an improved DQN-AC algorithm to optimize the long-term utility of the whole system.

**Table 1.** The comparative analysis of different work ("+": involved; "−": not involved).

| Ref. | Constraints | | | Infrastructure | | | Method |
|---|---|---|---|---|---|---|---|
| | **Time Delay** | **Energy Consumption** | **Multiple Users** | **Multiple Edge Servers** | **Cloud Server** | | |
| [15] | + | + | + | + | − | lyapunov optimization |
| [16] | + | + | + | − | − | deep Reinforcement learning |
| [18] | + | − | + | + | + | iterative heuristic |
| [21] | + | + | + | − | − | iterative algorithm |
| [22] | + | + | − | − | − | convex approximation |
| [24] | − | + | + | − | − | double deep Q network |
| [25] | − | + | + | + | + | deep Reinforcement Learning |
| [27] | + | + | + | − | + | semidefinite relaxation approach |
| [29] | + | + | + | − | − | exact line search algorithm |
| [30] | + | + | + | + | − | deep reinforcement learning |
| this paper | + | + | + | + | − | deep reinforcement learning |

## 3. Problem Formulation

The major abbreviations and symbols used in this paper are defined in Tables 2 and 3, respectively.

**Table 2.** List of abbreviations.

| Abbreviation | Description |
|---|---|
| MEC | Mobile Edge Computing |
| UDN | Ultra-Dense Network |
| MUD | Mobile User Device |
| DRL | Deep Reinforcement Learning |
| DQN | Deep Q Network |
| AC | Action Classification |
| DQN-AC | Deep Q Network with Action Classification |
| FLC | Full Local Computation |
| FOC | Full Offloading Computation |

**Table 3.** Symbol definitions.

| Symbol | Definition |
|---|---|
| $N$ | the set of all MUDs |
| $S$ | the set of all MEC servers |
| $a_{n,s}$ | whether MUD $n$ chooses MEC server $s$ for computation offloading |
| $r_{n,s}$ | the data transmission rate of MUD $n$ accessing to MEC server $s$ |
| $W$ | the wireless channel bandwidth |
| $p_{n,s}$ | the transmission power |
| $g_{n,s}$ | the channel gain |
| $A_d$ | the antenna gain |
| $f_c$ | the carrier frequency |
| $l_{n,s}$ | the distance between MUD $n$ and MEC server $s$ |
| $\xi$ | the path loss exponent |
| $\sigma^2$ | the white Gaussian noise |
| $R_n$ | the intensive task of MUD $n$ |
| $B_n$ | the data size of task $R_n$ |
| $D_n$ | the total number of CPU cycles required for completing the task |

**Table 3.** *Cont.*

| Symbol | Definition |
|---|---|
| $T_n^{\max}$ | the maximum delay for computing task $R_n$ |
| $T_n^l, T_{n,s}^e$ | local or edge execution delay |
| $E_n^l, T_{n,s}^e$ | local or edge energy consumption |
| $C_n^l, C_{n,s}^e$ | the weighted cost of local computing or edge computing |
| $f_n^l$ | the local computing power of MUD $n$ |
| $z_n$ | energy consumption density |
| $\theta_1, \theta_2$ | the weight parameters of execution delay and energy consumption |

### 3.1. System Model

As shown in Figure 1, the system model under the UDN scenario consists of multiple small base stations and multiple MUDs. Each small base station is equipped with a MEC server, named a MEC small base station. All MEC small base stations cover their service areas in an overlapping manner. The set of all MUDs and MEC servers is defined as $\mathbf{N} = \{1, 2, \dots, N\}$ and $\mathbf{S} = \{1, 2, \dots, S\}$. In the UDN scenario, it is assumed that each MUD has a compute-intensive task, and all computing tasks can be offloaded to one MEC server through the wireless channel. Due to the ultra-dense coverage of small base stations, MUDs will be in the service areas of multiple small base stations. To achieve the minimum system cost, small base stations communicate with each other to determine who performs the computing tasks, and then transmit it to the corresponding MEC server for processing.

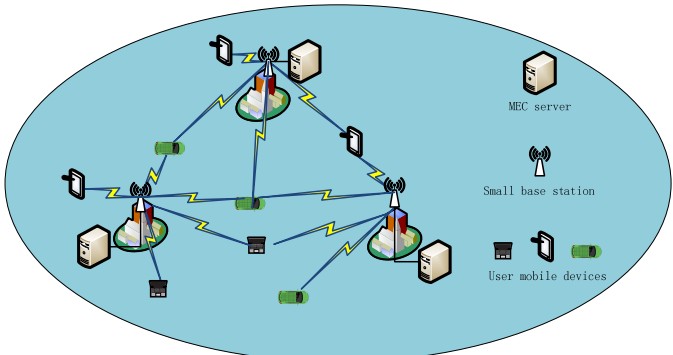

**Figure 1.** The system model under ultra-dense network scenario.

### 3.2. Communication Model

The offloading decision variable is defined as $a_{n,s} \in \{0, 1\}$, indicating whether MUD $n$ chooses the MEC server $s$ for computation offloading. If the MUD $n$ chooses to offload, the data transmission rate of access to the MEC server $s$ can be expressed as:

$$r_{n,s} = a_n W \log_2 \left(1 + \frac{p_{n,s} g_{n,s}}{\sigma^2 + \sum_{k=1, k \neq s}^{S} \sum_{j=1, j \neq n}^{N} p_{j,s} g_{j,s}}\right) \tag{1}$$

where $a_n = \sum_{s=1}^{S} a_{n,s} = \{0, 1\}$; $W$ is the wireless channel bandwidth; $p_{n,s}$ represents the transmission power of the MUD $n$ for uploading data; $\sigma^2$ is white Gaussian noise during data transmission; and $g_{n,s}$ is the channel gain in the wireless channel, which is expressed by (2).

$$g_{n,s} = A_d \left(\frac{3 \cdot 10^8}{4 \pi f_c l_{n,s}}\right)^{\xi} \tag{2}$$

where $A_d$ denotes the antenna gain, $f_c$ denotes the carrier frequency, and $l_{n,s}$ is the distance between MUD $n$ and MEC server $s$ and $\xi$ is the path loss exponent [16].

### 3.3. Computation Model

Assume the intensive task of MUD $n$ is $R_n = \{B_n, D_n, T_n^{\max}\}$, where $B_n$ is the data size, $D_n$ is the total number of CPU cycles required for completing the task, and $T_n^{\max}$ indicates the maximum delay for computing task $R_n$; that is, the task latency of each MUD cannot exceed $T_n^{\max}$. The task can be executed using a local computing model or edge computing model, which are introduced in the following sections.

### 3.3.1. The Local Computing Model

If the MUD $n$ chooses to perform $R_n$ locally, the cost includes local execution delay $T_n^l$ and energy consumption $E_n^l$. $f_n^l$ is defined as the local computing power of the MUD $n$, which is expressed by the CPU cycles per second.

The local execution delay is:

$$T_n^l = \frac{D_n}{f_n^l} \tag{3}$$

The local energy consumption is:

$$E_n^l = z_n \left(f_n^l\right)^2 D_n \tag{4}$$

In the above equation, $z_n$ is energy consumption density, and specific values can be obtained according to [34]. The parameter gaps of $z_n$ between different equipment kinds are very small. The weighted cost of local computing is:

$$C_n^l = \theta_1 T_n^l + \theta_2 E_n^l \tag{5}$$

where $0 \leq \theta_1, \theta_2 \leq 1$ represent the weight parameters of execution delay and energy consumption, respectively.

### 3.3.2. The Local Computing Model

The MUD $n$ chooses to perform the task through a MEC small base station. The whole execution process includes three parts: First, the MUD $n$ needs to upload data to the small base station $s$, then from the small base station to the MEC server. Second, the MEC server allocates certain computing resources to perform the task. Finally, the MEC server returns the result to the MUD $n$.

According to the above process, the first part is the transmission delay, which is expressed as:

$$T_{n,s}^u = \frac{B_n}{r_{n,s}} \tag{6}$$

The energy consumption corresponding to the first part is:

$$E_{n,s}^u = p_{n,s} T_{n,s}^u = \frac{p_{n,s} B_n}{r_{n,s}} \tag{7}$$

The second part is the processing delay of the MEC server. $f_{n,s}$ is the computing resources allocated by the MEC server $s$ for MUD $n$. The processing delay can be expressed as:

$$T_{n,s}^c = \frac{D_n}{f_{n,s}} \tag{8}$$

The MUD $n$ waits while the MEC server performs the task. The idle power of the mobile user's device in this state is set to $P_n^w$; then, the energy consumption during this period is:

$$E_{n,s}^w = P_n^w T_{n,s}^c = \frac{P_n^w D_n}{f_{n,s}} \tag{9}$$

For the last part, according to [35], the return rate of the wireless network is generally much higher than that of the offloaded data, and the execution result is much smaller

than that of the input data. The execution delay and energy consumption are, therefore, generally ignored. The execution delay and energy consumption are, respectively:

$$T_{n,s}^e = T_{n,s}^u + T_{n,s}^c = \frac{B_n}{r_{n,s}} + \frac{D_n}{f_{n,s}} \tag{10}$$

$$E_{n,s}^e = E_{n,s}^e + E_{n,s}^w = \frac{p_{n,s} B_n}{r_{n,s}} + \frac{P_n^w D_n}{f_{n,s}} \tag{11}$$

In sum, the weighted cost of edge computing is:

$$C_{n,s}^e = \theta_1 T_{n,s}^e + \theta_2 E_{n,s}^e \tag{12}$$

According to Equations (2)–(11), the weighted cost of all users can be obtained; namely, the system objective function is:

$$C_{all} = \sum_{n=1}^N \left\{ (1 - \sum_{s \in S} a_{n,s}) C_n^l + \sum_{s \in S} a_{n,s} C_{n,s}^e \right\} \tag{13}$$

*3.4. Problem Formulation*

To minimize the total system cost, it is necessary to find the best offloading decision and resource allocation scheme. The problem is described as follows:

$$\begin{aligned}
&\min_{(A,f)} C_{all}(A, f) = \min_{(A,f)} \sum_{n=1}^N \left\{ (1 - \sum_{s \in S} a_{n,s}) C_n^l + \sum_{s \in S} a_{n,s} C_{n,s}^e \right\} \\
&s.t.\ C_1 : a_{n,s} \in \{0,1\}, \forall n \in \mathbf{N}, \forall s \in \mathbf{S} \\
&\quad\ C_2 : \sum_{s \in S} a_{n,s} \le 1, \forall s \in \mathbf{S} \\
&\quad\ C_3 : T_n^l, T_{n,s}^e \le T_n^{\max}, \forall n \in \mathbf{N}, \forall s \in \mathbf{S} \\
&\quad\ C_4 : f_{n,s} \ge 0, \forall n \in \mathbf{N}, \forall s \in \mathbf{S} \\
&\quad\ C_5 : \sum_{s=1}^S f_{n,s} \le f_s^{\max}, \forall s \in \mathbf{S}
\end{aligned} \tag{14}$$

In Equation (14), $A = \{a_1, a_2, \ldots, a_N\}$ is the offloading decision vector, and $f = \{f_1, f_2, \ldots, f_N\}$ is the resource allocation vector. $C_1$ and $C_2$ indicate that each MUD performs the task only by local computing or by edge computing, respectively. $C_3$ means neither local computing delay nor edge computing delay can exceed the maximum tolerance delay $T_n^{\max}$. $C_4$ and $C_5$ indicate that the computing resources allocated to MUDs are non-negative, and the total allocated resources cannot exceed $f_s^{\max}$.

The reasons that the optimization function is difficult to solve directly are as follows: Equation (14) is a mixed integer nonlinear programming problem. The existence of binary variables makes them nonconvex functions, which cannot be solved via the conventional solution of convex optimization. At the same time, the complexity of the optimization function is too high. If the two optimization variables ($A = \{a_1, a_2, \ldots, a_N\}$ and $f = \{f_1, f_2, \ldots, f_N\}$) are binary variables, the complexity of the original optimization problem is O ($N^2$).

## 4. Proposed Method

Based on the above problem model, DRL was adopted to solve the problem. On the one hand, reinforcement learning allows agents to obtain rewards in the process of interaction with the environment in a "trial and error" manner to guide behavior and improve decision making, which is suitable for the joint optimization of offloading decision and resource allocation in this model. On the other hand, deep learning can avoid the storage difficulties caused by excessive state space and action space. In this paper, we utilized the deep Q network (DQN), which is a typical DRL algorithm, to solve the problem. Combined with the problem model, the three elements of DQN are defined in detail:

state, action and reward. Then, an action classification (AC) algorithm was proposed to filter effective execution actions, which partially improved the DQN, named the DQN-AC algorithm. The offloading decision and resource allocation scheme-based DQN-AC algorithm was proposed to minimize the objective function.

### 4.1. The Definition of State, Action and Reward

DQN is made up of a deep neural network and value-based Q-learning algorithm. Q is $Q(S, a)$, which represents the expectation that action $a$ can be selected under the state $S$ at a certain moment. The environment responds to the agent's actions with a reward $R(S, a)$.

The system state $S$ consists of two parts $S = (X, Y)$. $X = C_{all}$ represents the system cost. $Y = \{y_1, y_2, \ldots, y_S\}$ indicates the idle resource on each MEC server. We can obtain $y_s$ from (15).

$$y_s = f_s^{max} - \sum_{n=1}^{N} f_{n,s}, s \in \mathbf{S} \tag{15}$$

The system action is defined as $a = \{a_1, a_2, \ldots, a_N, f_1, f_2, \ldots, f_N\}$, which combines the offloading decision vector $A = \{a_1, a_2, \ldots, a_N\}$ and the resource allocation vector $f = \{f_1, f_2, \ldots, f_N\}$. The reward function is set to $R(S, a)$ which is expressed by (16).

$$R(S, a) = \frac{X_{local} - X(S, a)}{X_{local}} \tag{16}$$

The larger the $R(S, a)$, the smaller the $X(S, a)$ in the current state.

### 4.2. Action Classification Algorithm

According to the constraint conditions $C_4$ and $C_5$, the AC algorithm was added to the action selection part of the DQN algorithm to filter effective actions and improve learning efficiency. The process of AC algorithm is described in Algorithm 1:

---
**Algorithm 1** AC algorithm

---
input $s_t, a_t$
output bool //reasonable judgment of Boolean value by action
initialization bool = False
if $a_{n,s} = 0$ // UM chooses to perform calculations locally
  if $T_n^l \leq T_n^{max}$ // whether local computing latency constraints are met
    bool = True // action allows execution
    else:
        $a_{n,s} = 1$
if $a_{n,s} = 1$ //offloading computation
  if $s = j$ // select the $j$-th MEC for offloading computation
      if $T_{n,s}^e > T_n^{max}$
      bool = False

      elif $T_{n,s}^e \leq T_n^{max}$ and $\sum\limits_{n=1}^{N} f_{n,s} > F_s$:
      bool = False
      else
         $s! = j$ //select new MEC small base station
  else

      if $T_{n,s}^e \leq T_n^{max}$ and $\sum\limits_{n=1}^{N} f_{n,s} \leq F_s$:
      bool = True
  end if

---

### 4.3. DQN-AC Algorithm

The main aim of Q-learning is to build a Q-table of states and actions, and select the action that can obtain the maximum reward according to the Q value. Q-learning needs to

calculate each state–action group and store its corresponding Q value in the table. A deep neural network was introduced to solve the dimension disaster problem of Q-learning. The state and action were taken as the input of the neural network, and then the Q value was obtained after the analysis of the neural network. Then, the AC algorithm was combined with the DQN algorithm. When meeting the execution delay and resource constraints, $a_t$ was performed. Otherwise, the action was selected based on the greedy policy. The specific implementation process is described in Algorithm 2:

---

**Algorithm 2** DQN-AC algorithm

---

initialize replay memory $D$ to capacity $N$
initialize $Q, \theta, Q', \theta'$
for $episode = 1, M$ do
initialize sequence $s_1 = \{x_1\}$ and preprocessed $\phi_1 = \phi(x_1)$
for $t = 1, T$ do
  if $rand() > \varepsilon$ then
    $a_t = rand(a)$;
  else
    $a_t = \underset{a \in A}{\mathrm{argmax}} Q(\phi(s_t), a|\theta)$
  end if
  if $AC(s_t, a_t)$ then //filtering actions using AF algorithm
    $s_{t+1} = s_t, a_t, x_{t+1}$;
    $\phi_{t+1} = \phi(s_{t+1})$;
    store transition $(\phi_t, a_t, r_t, \phi_{t+1})$ in $D$
  if $t \equiv 0 \bmod K$ then
    sample random minibatch of transitions $(\phi_j, a_j, r_j, \phi_{j+1})$ from $D$
    set $y_j = \begin{cases} r_j & \text{for terminal } \phi_{j+1} \\ r_j + \gamma \underset{a' \in A}{\max} Q'(\phi_{j+1}, a' \mid \theta') & \text{for non-terminal } \phi_{j+1} \end{cases}$
    perform a gradient descent step on $\Delta\theta = (y_j - Q(\phi_j, a_j|\theta))^2$
    update $\theta = \theta + \Delta\theta$
  end if
  update the network weight every C steps: $\theta' = \theta$
  end for
end for

---

*4.4. The Performance Evaluation of DQN-AC*

The computational complexity analysis of the DQN-AC was evaluated as follows. There are $S$ MEC servers and $N$ MUDs. The total caching capacity of all MEC servers is $F = \sum_{s=1}^{S} f_s^{\max}$, and the total size of all intensive tasks in MUDs is $B = \sum_{n=1}^{N} B_n$. The number of computation offloading strategies for $N$ MUDs and resource allocation decision for $S$ MEC servers are $2^N$ and $2^S$, respectively. Thus, the complexity of the exhaustive search for the optimal solution is $O(2^{N+S} N^{B+S})$, which is an extremely difficult task. For the proposed DQN-AC algorithm, the computational complexity is $O(N \cdot S \cdot F \cdot B)$. Consequently, the proposed algorithm holds lower computational complexity than the exhaustive search. Additionally, if there exists one more states in the final policy, the proposed algorithm will keep updating until the state does not change, which clearly shows that the policy is not the final one. Therefore, the proposed algorithm is stable.

**5. Experimental Results and Analysis**

This paper evaluated the performance of the proposed DQN-AC algorithm through the Python platform, compared with the full local computation (FLC) algorithm, which indicates that all users select local computing; the full offloading computation (FOC) algorithm, which indicates that all users choose edge computing; and the Q-learning algorithm. Assume that 20 MEC small base stations cover an area of 300 m$^2$, and there are 60 MUDs in the area. The size of the computing task for each user is randomly distributed

between 300 and 500, the number of computing resources required by each user is randomly distributed between 900 and 1100, and the computing resource of the MEC server is evenly allocated to each user. Detailed simulation parameters are shown in Table 4.

**Table 4.** The system simulation parameters.

| Parameter Description | Parameter Value Domain |
|---|---|
| wireless channel bandwidth $W$ | 10 MHz |
| thermal noise of wireless environment system $\sigma^2$ | $-100$ dBm |
| the path fading factor $\xi$ | 3 |
| the antenna gain $A_d$ | 4 |
| the carrier frequency $f_c$ | 915 MHz |
| UMD transmission power $p_{n,s}$ | 100 mw |
| UMD idle power $P_n^w$ | 10 mw |
| the size of input data $B_n$ | 300 Kb–500 Kb |
| MEC computing capability $f_s^{\max}$ | 20 GHz/s |
| UMD computing ability $f_n^l$ | 1 GHz/s |
| number of computing resources $D_n$ | 900 hz–1100 hz |
| maximum tolerance delay $T_n^{\max}$ | $3 \times 10^{-3}$ s |
| weight $\theta_1, \theta_2$ | 0.5, 0.5 |

Figure 2 shows the relationship between the system weighted cost and the number of MUDs with 20 MEC small base stations. As the number of MUDs increased, all curves showed an upward trend. For the same number, the DQN-AC algorithm had the minimum system weighted cost. When the number of MUDs was less than 20, the system weighted costs of the other three algorithms were similar, except for the FLC algorithm. In this case, the computation resources of the MEC servers were sufficient, and the users were more inclined to conduct edge computing. With the further increase in the number of MUDs, which was close to 80, there was not much difference between the FOC and Q-learning algorithms, but there was a big gap between the two algorithms and the DQN-AC algorithm. When the number of MUDs was larger than 80, the gap between the FOC and Q-learning algorithms gradually increased. However, the performance of the DQN-AC was still stable and had the best effect. This is because edge computing users compete with each other for the limited resources in MEC servers. In the FOC algorithm, all users chose to compute the task in the MEC server, which led to the rapid increase in the system weighted cost. The DQN-AC algorithm made full use of local terminal resources, reduced competition among users and allocated computing resources reasonably.

Figure 3 shows the curve of system weighted cost with the number of MEC small base stations for 30 MUDs. The curve of FLC algorithm was almost unchanged because the task was executed in a local terminal, which has nothing to do with MEC small base stations. When the number of MEC small base stations was small, there was a small gap between the FOC, Q-learning and DQN-AC algorithms. This is because there were few options for offloading. As the number of MEC small base stations increased, the FOC, Q-learning and DQN-AC algorithms all showed a downward trend. They had a greater chance to select the best MEC small base station for offloading, so the system weighted costs were gradually reduced. However, when the number of MEC small base stations was approximately over 30, the curves of these three algorithms converged gradually, because when the number of MEC small base stations reaches a certain amount, the optimal MEC small base station they choose will not change anymore. It can be seen from the figure that the curve of the DQN-AC algorithm was always at the bottom, and could achieve the best effect.

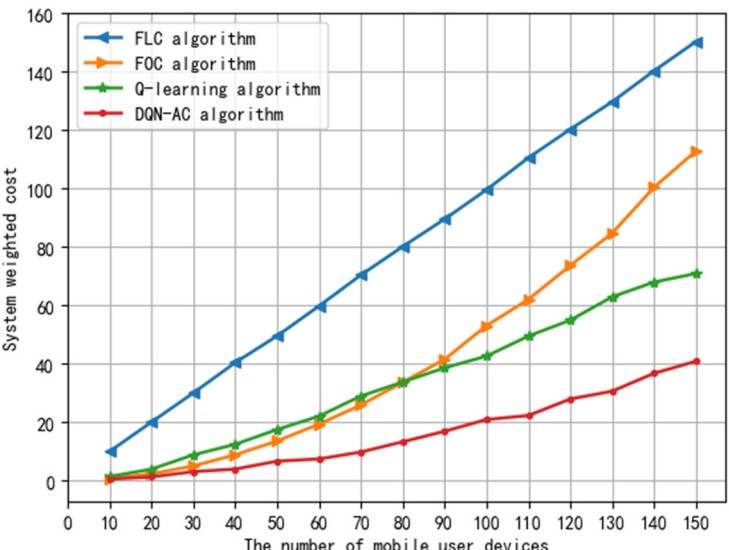

**Figure 2.** The impact of the number of mobile user devices on system weighted cost.

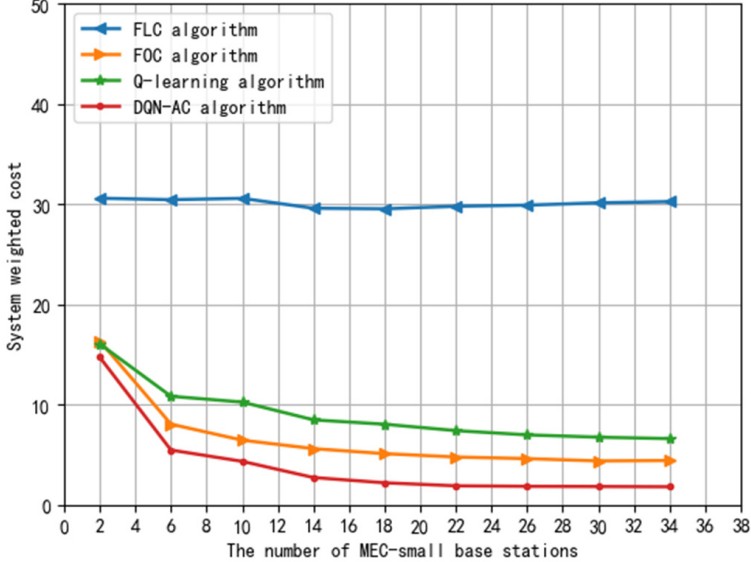

**Figure 3.** The impact of the number of MEC small base stations on system weighted cost.

Figure 4 shows a diagram of the system weighted cost versus the capacity of the MEC server for 50 MUDs and 20 MEC small base stations (assume that the capacity of all base stations is the same). With the increase in the capacity of the MEC server, the curves of the other three algorithms showed a downward trend, except the FLC algorithm. The curve of the FLC algorithm was almost unchanged, because the capacity of the MEC server did not affect the local computing process. While for edge computing, users could allocate enough resources as the capacity increased, thus reducing delay and energy consumption, it can be seen that the DQN-AC algorithm had the best effect. The changing trend of the curve shows that it is not easy to complete offloading computation when the capacity of MEC server is small. However, when computing resources were sufficient, the system weighted costs changed slightly, which is because there were extra resources unused by Q-learning.

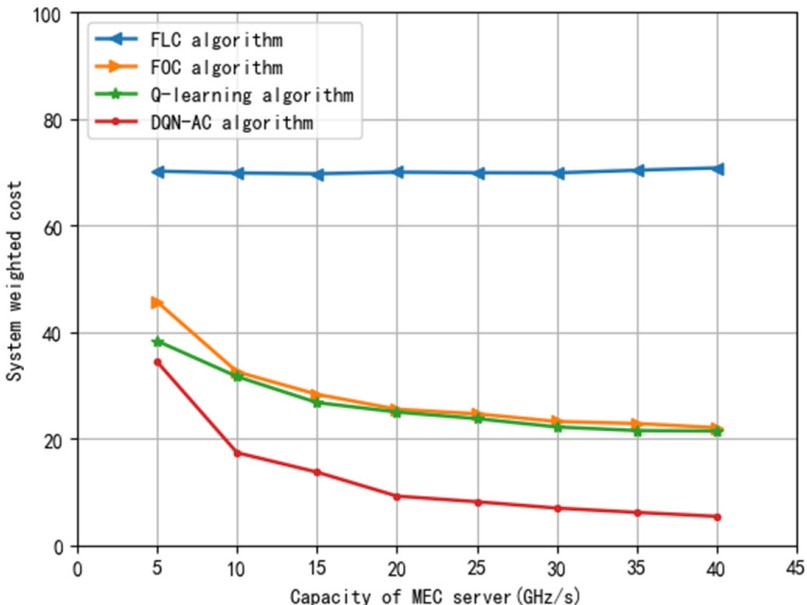

**Figure 4.** The impact of MEC server capacity on system weighted cost.

As shown in Figures 5 and 6, all four methods showed an upward trend as the horizontal axis increased. The larger amount of data/CPU cycles required more time to transmit, which also increased the energy consumption. With the increase in input data/CPU cycles, the gap between the FLC algorithm and the other three algorithms became larger. The system weighted cost was always higher than those of the Q-learning and DQN-AC algorithms. It is difficult to complete complex tasks using local computing, which results in the continuous increase in the system weighted cost. It can be seen that the DQN-AC algorithm had the best effect and the slowest rise compared with the other three algorithms.

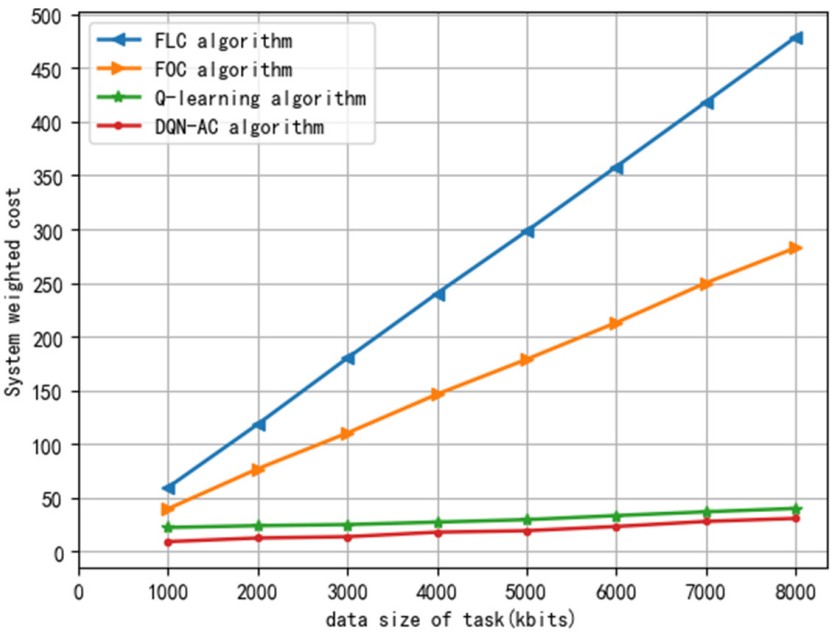

**Figure 5.** The impact of data size of task on system weighted cost.

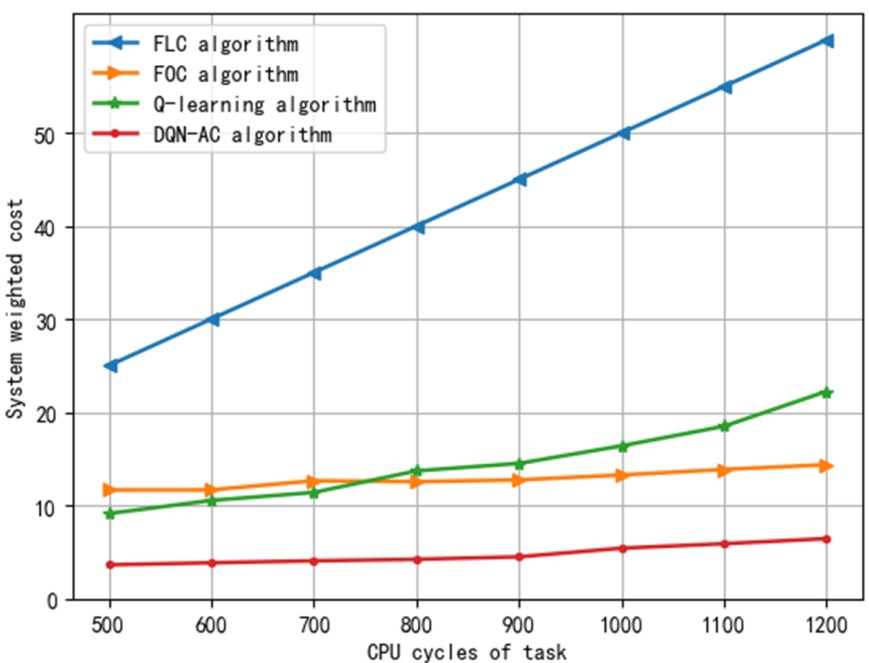

**Figure 6.** The impact of CPU cycles of task on system weighted cost.

## 6. Conclusions

In this article, we first designed a MEC architecture based on a UDN; then established a system weighted cost based on execution delay and energy consumption; and finally proposed an offloading decision and resource allocation scheme-based DQN-AC, which can balance the near optimal system utility and computational complexity. The simulation results validate the effectiveness of the DQN-AC and demonstrate that the DQN-AC outperforms the FLC, FOC and Q-learning algorithms.

In our future work, with the rapid development of code decomposition and parallel computing, we will consider more complicated scenarios, such as partial offloading, bandwidth fluctuation and server failure. Moreover, our future work will further investigate the problem of offloading security in wireless networks.

**Author Contributions:** Conceptualization, C.D. and P.D.; methodology, P.D. and Q.G.; validation, B.L. and Q.G.; formal analysis, C.D. and Q.G.; investigation, B.L.; data curation, P.D.; writing—original draft preparation, P.D.; writing—review and editing, C.D. and Q.G.; visualization, B.L.; supervision, Y.L. All authors have read and agreed to the published version of the manuscript.

**Funding:** This research was funded by the National Natural Science Foundation of China (grant numbers 61775057 and 61971190), the Key Project of Science and Technology Research in Higher Education of Hebei Province (grant number ZD2021406) and the Fundamental Research Funds for the Central Universities (grants number 2021MS086).

**Institutional Review Board Statement:** Not applicable.

**Informed Consent Statement:** Not applicable.

**Data Availability Statement:** Not applicable.

**Conflicts of Interest:** The authors declare that they have no conflict of interest or any personal circumstances that may be perceived as inappropriately influencing the representation or interpretation of reported research results.

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
