# Peer review of "MEC Computation Offloading-Based Learning Strategy in Ultra-Dense Networks"

_information, doi:10.3390/info13060271_

Round 1

Reviewer 1 Report

The paper is very interesting and it is within the scope of the information Journal. However, some important improvements are required. My comments are below.

  1. Please avoid the usage of lumped references. It is important to describe clearly which is the contribution of each one of the references used.
  2. In the section named related works, it is important to include a table that summarizes the main literature reports to make evidence of the main contribution of this manuscript.
  3. The mathematical optimization model was clearly developed. However, I recommend including a complete nomenclature list at the end of the document following the Journal rules with all the variables, parameters, and main acronyms.
  4. Model characteristics in a set of Equations (13) must be well described, with its geometrical characteristics and challenges to be addressed. 
  5. Conclusions must be rewritten including the main numerical achievements. In addition, some possible future works will help continue developing this research area.

Reviewer 2 Report

The manuscript concerns the problem of development the strategy for operation of cooperative computation mobile user devices with limited resources. Certainly this problem is complex problem which requires remote cloud, storage facilities coupled with providing stable anf reliable network  infrastructure. The authors have proposed an interesting approach to offload intensive computation tasks to reduce the computation time delays.
Strong sides of the research are given below:
1) math problem formulation;
2) using the DQN-AC algorithm;
3) performance evaluation of the proposed approach.
In my opinion there are several comments could improve the readability of the research.
1) Introduction requires more clear problem formulation and the contribution of the proposed approach.
2) Several math definition require the clarification of abbreviates. For example, in line 136 and further in equation (13).
3) It is better to clarify what are MUD's  and give descriptive examples of those.
4) The performance evaluation of DQN-AC which is presented in section 4 can be explained in the following section.
5) It is better to extend the conclusion. This section should fully reflect the fulfilled research and support the main findings.

However I can recommend this manuscript as a candidate for publishing without essential revision. 

Round 2

Reviewer 1 Report

The authors have addressed in a positive manner all of my comments. The paper has been largely improved. In my opinion, the article is now suitable for publication in its current form.
